# Never Change a Flowing System? The Effects of Retrograde Flow on Isolated Perfused Lungs and Vessels

**DOI:** 10.3390/cells10051210

**Published:** 2021-05-15

**Authors:** Hanif Krabbe, Sergej Klassen, Johannes Bleidorn, Michael J. Jacobs, Julia Krabbe, Aaron Babendreyer, Christian Martin

**Affiliations:** 1Institute of Pharmacology and Toxicology, Medical Faculty, RWTH Aachen University, Wendlingweg 2, 52074 Aachen, Germany; klassen.ser@gmail.com (S.K.); johannes.bleidorn@rwth-aachen.de (J.B.); chmartin@ukaachen.de (C.M.); 2European Vascular Centre Aachen-Maastricht, Department of Vascular Surgery, Medical Faculty, University Hospital RWTH Aachen, Pauwelsstraße 30, 52074 Aachen, Germany; mjacobs@ukaachen.de; 3Institute of Occupational, Social and Environmental Medicine, Medical Faculty, RWTH Aachen University, Pauwelsstraße 30, 52074 Aachen, Germany; jkrabbe@ukaachen.de; 4Institute of Molecular Pharmacology, Medical Faculty, RWTH Aachen University, Wendlingweg 2, 52074 Aachen, Germany; ababendreyer@ukaachen.de

**Keywords:** retrograde perfusion, isolated perfused vessel, isolated perfused lungs, flow reversal, retrograde flow

## Abstract

Retrograde perfusion may occur during disease, surgery or extracorporeal circulation. While it is clear that endothelial cells sense and respond to changes in blood flow, the consequences of retrograde perfusion are only poorly defined. Similar to shear stress or disturbed flow, retrograde perfusion might result in vasomotor responses, edema formation or inflammation in and around vessels. In this study we investigated in rats the effects of retrograde perfusion in isolated systemic vessels (IPV) and in pulmonary vessels of isolated perfused lungs (IPL). Anterograde and retrograde perfusion was performed for 480 min in IPV and for 180 min in the IPL. Perfusion pressure, cytokine levels in perfusate and bronchoalveolar lavage fluid (BALF), edema formation and mRNA expression were studied. In IPV, an increased perfusion pressure and initially also increased cytokine levels were observed during retrograde perfusion. In the IPL, increased edema formation occurred, while cytokine levels were not increased, though dilution of cytokines in BALF due to pulmonary edema cannot be excluded. In conclusion, effects of flow reversal were visible immediately after initiation of retrograde perfusion. Pulmonary edema formation was the only effect of the 3 h retrograde perfusion. Therefore, further research should focus on identification of possible long-term complications of flow reversal.

## 1. Introduction

In human patients, flow direction changes in systemic vessels occur under several pathological conditions and iatrogenic procedures. Arterial blood flow reversal can be observed in the case of arterial stenoses with post stenotic pressure drops leading to a consecutive retrograde flow in the off-branches of these arteries. The subclavian steal syndrome is a frequent example of this phenomenon with retrograde perfusion in vertebral arteries [1]. Retrograde flow can also lead to a serious complication after endovascular arterial repairs, i.e., type 2 endoleak after stent implantation caused by retrograde flow in side branches of arteries. This can reactivate the aneurysms, finally nullifying the whole purpose of the surgery [2]. In children with the Bland–White–Garland syndrome flow reversal occurs in coronary arteries [3]. Therapeutically, cardio-pulmonary bypass and extracorporeal membrane oxygenation during vascular or heart surgery as well as therapy of combined heart and lung failure, in case of peripheral arteriovenous cannulation, are all based on a constant retrograde flow, in order to achieve coronary and/or cerebral perfusion through the aorta for a time frame stretching from several hours up to many days.

In the lungs, retrograde perfusion of the pulmonary circulation only occurs as a rare ultima ratio therapy of pulmonary embolisms as intraoperative flow reversal to wash out remaining thrombotic material after thrombectomy. In experiments, retrograde lung perfusion was examined in very few experimental studies in rat [4] and mouse [5] isolated perfused lungs (IPL) for short periods of time (60 min) and with high failure rates. Interestingly, in retrogradely perfused mouse IPL, a large amount of edema formation occurred when the lungs were also damaged by overventilation [5]. 

Endothelial cells can detect flow direction relative to their morphological and cytoskeletal axis [6,7] and change their polarization in adaptation to shear stress [7,8,9]. Changes in flow direction cause induction of several, even opposed pathways in endothelial cells. One study observed how change in flow direction led to the induction of endothelial nitric oxide synthase (eNOS) and the pro-inflammatory transcription factor nuclear factor κ-light-chain-enhancer of activated B-cells (NFκB) [6]. Induction or lack of eNOS have been shown to alter the functionality of tight junctions and result in hyperpermeability with consecutive pulmonary edema [10,11]. Another study identified the positioning of the nucleus in the cell architecture as an endothelial cell flow direction sensor. Elastic stress in the nucleus indicates shear stress with effects on the actin cytoskeleton and on cell–cell adhesions [12]. However, those experiments were conducted in mono cell cultures in vitro, where only one cell type was cultured on unphysiologically stiff surfaces [13,14]. Since pulmonary vessels are well known to respond differently in vitro and in situ [15], critical effects of flow reversal need to be studied in intact tissue.

Taken together, the effects of retrograde perfusion on vessels in situ are not well defined. The scarce literature of retrograde perfusion suggests vasomotor effects in systemic and pulmonary vessels, e.g., via eNOS induction, edema formation in the lungs and increased inflammation in or around vessels. In this study we investigated the effects of retrograde perfusion in isolated vessels (IPV) and lungs (IPL) of rats for 8 h or 3 h, respectively. Our hypothesis was that retrograde perfusion would lead to three effects: (1) alteration of vasotonus, (2) increased edema formation in the lungs, and (3) induction of inflammation. Therefore, perfusion pressure in IPV and inflow in IPL were compared for anterograde and retrograde perfusion, as well as the induction of genes relating to vasotonus in IPL. Edema formation derived from lung weight and wet-to-dry ratio was studied in IPL and cytokine levels were determined in perfusate samples and bronchoalveolar lavage fluid (BALF).

## 2. Materials and Methods

### 2.1. Animals

Female Wistar rats (~300 g) obtained from Janvier (Le Genest-Saint-Isle, France) were used as lung or vessel donors for IPL. They were randomly assigned to one of the groups. 

All mice were housed in an individually ventilated cage system (Positive/Negative Control IVC; Allentown Inc., Allentown NJ, USA) according to the recommendations of the Federation of Laboratory Animal Science Associations (FELASA) and German Society of Laboratory Animal Science (GV-SOLAS).

### 2.2. Agents

The buffer supplements were all obtained from Merck (Darmstadt, Germany), except sodium chloride, potassium chloride, d-(+)-glucose monohydrate, calcium nitrate tetrahydrate, MEM Amino Acids (50×), MEM Non Essential Amino Acids (100×), and MEM Vitamins (100×), which were from Sigma-Aldrich (Steinheim, Germany), and Glutathione, Dulbecco’s Phosphate-Buffered Saline (DPBS) and Ultraglutamine, which were purchased from Lonza (Basel, Switzerland). Pentobarbital (Narcoren) was purchased from Merial (Hallbergmoos, Germany).

### 2.3. Isolated Perfused Rat Vessel Preparation (IPV)

IPVs were prepared from female rats. After sacrificing the rat via peritoneal injection of sodium pentobarbital (60 mg/kg) and cutting through the renal aorta, the thoraco-abdominal aorta was removed. Macroscopic branch-off vessels of the aorta were ligated and smaller vessels, e.g., spinal arteries, were closed with thermal coagulation. Afterwards, the vessel was placed in a modified chamber from mouse IPL (Hugo Sachs Elektronik, March Hugstetten, Germany) (Figure 1) and the ends were connected to two perfusing cannulas fixed with ligatures. Perfusion buffer was added to the chamber to avoid drying out and the perfusion was established with a step-wise rise of flow up to 30 mL/min. Hydrostatic resistance was adjusted to a perfusion pressure of 45 cm H_2_O at the outflow by elevating the perfusion buffer reservoir. Afterwards the hydrostatic resistance and flow were fixed and perfusion pressure was constantly monitored. The perfusing buffer consisted of a minimal essential medium (MEM) containing CaCl_2_ (1.8 mM), MgSO_4_ (0.8 mM), KCl (5.4 mM), NaCl (116.4 mM), glucose (16.7 mM), NaHCO_3_ (26.1 mM), Hepes (25.17 mM), sodium pyruvate, amino acids, vitamins and glutamine as described before [16]. Every 30 min, 500 µL of the buffer was taken for sampling and replaced with fresh buffer.

Based on random allocation, the aorta was perfused via anterograde or retrograde perfusion. Retrograde perfusion was established just like the anterograde perfusion except that the inflow cannula was placed in the abdominal end of the aorta and the outflow cannula in the thoracic end. The experiment was conducted for 480 min. Data regarding perfusion pressure and flow were recorded and analyzed by the Pulmodyn software (Hugo Sachs Elektronik).

### 2.4. Isolated Perfused Rat Lung Preparation (IPL)

For IPL, rats were sacrificed in the same procedure as described for IPV. Afterwards, the lungs were prepared as described before [16] and ventilated with a respiratory rate of 70 breaths per minute, an end-inspiratory pressure of −8 cm H_2_O and an end-expiratory pressure of −2 cm H_2_O, resulting in a tidal volume of about 2 mL. Recirculating perfusion was performed anterogradely via truncus pulmonalis, cannulated through the right ventricle, and left ventricle or retrogradely via the left atrium and truncus pulmonalis (Figure 2) with a constant pressure of 10 cm H_2_O, resulting in a perfusing flow of ~30 mL/min. The perfusing buffer contained Krebs–Henseleit buffer with 2% very low endotoxin bovine serum albumin (BSA) and was added with various supplements including sodium phosphate and sodium hydrogen carbonate as described before [17].

Based on random allocation, rats received either anterograde or retrograde perfusion. Retrograde perfusion was established just like the anterograde perfusion except that the tubings of both cannulas were switched in the cover of the ventilation chamber. The experiment was conducted for 180 min. The lung weight was measured over time with an integrated dynamometer. All data were recorded and analyzed by the Pulmodyn software (Hugo Sachs Elektronik).

### 2.5. Cytokine Determination—Electrochemiluminescence Assay

Perfusate samples from IPV and IPL, as well as BALF from IPL were analyzed for cytokines. The perfusate samples were obtained every 60 min for IPV and every 30 min for IPL. BALF was collected from the left lung by lavage with 500 µL physiological saline solution (0.9%) immediately after the end of ventilation. Levels of interferon γ (IFN-γ), interleukin (IL-) 1β, IL-4, IL-5, IL-6, keratinocytes-derived chemokine (KC), IL-10, IL-13, and tumor necrosis factor α (TNF-α) were determined. The V-PLEX Proinflammatory Panel 2 (Rat) was used for electrochemiluminescence immunoassays according to the manufacturer’s instructions (Meso Scale Discovery (MSD), Gaithersburg, MD, USA) using the Meso Quick Plex SQ 120. Raw data were analyzed using the Discovery Workbench 4.0 software (MSD).

### 2.6. Lung Wet-to-Dry Ratio

After 180 min of ventilation, the wet weight of the right superior lobe was recorded (wet weight). After desiccation at 40 °C for 7 days the dry weight was determined.

### 2.7. Reverse Transcription Quantitative Polymerase Chain Reaction (RT-qPCR)

Quantification of mRNA levels was conducted via RT-qPCR with normalization of levels to the mRNA levels of different reference genes. To identify the most stable reference genes the geNorm algorithm included in the Bio-Rad CFX Maestro 1.1 (Version 4.1.2433.1219) was used on six selected genes. Based on these results, glyceraldehyde 3-phosphate dehydrogenase (Gapdh), ribosomal protein lateral stalk subunit P0 (Rplp0) and TATA-binding protein (Tbp) were chosen as reference genes. All three reference genes were used and referred to as Reference Gene Index. After the experiments, the aorta and left lung were snap frozen in liquid nitrogen and stored at −80 °C. Samples were then ground with a mortar over liquid nitrogen for RNA extraction. RNA extraction was performed using RNeasy Kit (Qiagen, Hilden, Germany) and quantified photometrically. After that, only RNA from lungs was transcribed since the vessel samples resulted in a very low amount of RNA to analyze gene expression. Reverse transcription with equal amounts of RNA was conducted with PrimeScript™ RT Reagent Kit (Takara Bio Europe, St-Germain-en-Laye, France) and PCR reactions were performed using iTaq Universal SYBR Green Supermix (Bio-Rad, Hercules, CA, USA) according to the manufacturer’s instructions. Specific primers and annealing temperatures can be found in Supplementary Table 1. All PCR reactions were run on a CFX Connect Real-Time PCR Detection System (Bio-Rad) with the following protocol: 40 cycles of 10 s denaturation at 95 °C, followed by 10 s annealing at the indicated temperature and 15 s amplification at 72 °C. PCR efficiency was determined from the uncorrected RFU values using LinRegPCR [17]. Relative quantification was performed with the CFX Maestro Software 1.1 (Bio-Rad) (Version 4.1.2433.1219).

### 2.8. Statistics

Based on data from Krabbe and colleagues [5] and our own preliminary data, the experiments were planned with a statistical power of 80% and an alpha error of 0.05 (corrected for multiple comparisons) in order to detect differences in wet-to-dry ratio greater than 1.25 (G*Power v. 3.1.9.2, Düsseldorf, Germany [18,19]), which resulted in a group size of at least *n* = 5. Data analysis was performed using SAS 9.4 (SAS Institute Inc., Cary, NC, USA). All data are shown as mean ± SEM and *n* indicates the number of animals. Analysis was carried out using general linear mixed model analysis (Proc Glimmix) assuming a normal distribution. Additionally, data was assumed to be derived from lognormal (cytokines) or beta (percentage data) distribution; residual plots and the Shapiro–Wilk test were used as diagnostics. In case of heteroscedasticity, the degrees of freedom were adjusted by the Kenward–Roger approximation. To analyze the differences between groups the LSMEANS statement was included. *P* values were always adjusted by the simulated-Shaffermethod. Regarding repeated measurements (Figures 3, 5 and 6B), data were reduced to measurements of every 60 min for IPV and every 30 min for IPL. Data were plotted with GraphPad Prism 6 (GraphPad, La Jolla, CA, USA). Analysis differences were assumed to be significant with *p* < 0.05.

## 3. Results

### 3.1. IPL and IPV—Experimental Results

#### 3.1.1. IPV—Perfusion Pressure and Cytokine Levels

Perfusion pressures were significantly different for perfusion direction throughout the experiments (Figure 3). Regarding cytokine levels, there was a significant increase of KC levels for anterograde and retrograde perfusion over time, but no significant differences between perfusion directions (Figure 4A). IL-6 levels showed significant differences for perfusion directions for the 60 min timepoint (Figure 4B). There were also significant differences for the 60 and 120 min timepoints in IL-1β levels (Figure 4C). Interestingly, for IL-5 levels a significant difference could be observed between anterograde and retrograde perfusion for 60, 120 and 180 min (Figure 4D). Levels of IFN-γ, IL-4, IL-10, IL-13, and TNF-α were under the detection limit.

#### 3.1.2. IPL—Tidal Volume, Airway Resistance, Inflow and Perfusion Pressure

There were no significant differences for perfusion direction regarding inflow (Figure 5A) or perfusion pressure (Figure 5B), as well as tidal volume (Appendix A) and airway resistance (Appendix A).

#### 3.1.3. IPL—Edema Formation Indicated by Wet-to-Dry Ratio and Lung Weight

Regarding lung edema formation, significant differences between anterograde and retrograde perfusion could be observed for wet-to-dry ratios (Figure 6A). Accordingly, the lung weight was significantly different between anterograde and retrograde perfusion direction from 30 min until the end of the experiments (Figure 6B).

#### 3.1.4. IPL—Cytokine Levels in Perfusate and BALF

In contrast to IPV, IPL perfusate cytokine levels showed no significant differences between anterograde and retrograde perfusion (Figure 7). An increase depending on incubation time could be observed for IL-6, KC and TNF-α (Figure 7C–E), in contrast IFN-γ, IL-1β and IL-5 were relatively constant (Figure 7A,B,F). IFN-γ and IL-5 tended to be slightly higher in retrograde perfusion. Levels of IL-4, IL-10 and IL-13 were under the detection limit. Similar to perfusate, no significant differences could be detected for perfusion direction for BALF (Figure 8). Levels of IL-4, IL-10 and IL-13 were under the detection limit.

#### 3.1.5. IPL—mRNA Expression Levels

Similar to cytokine levels in perfusate and BALF, no significant differences could be found in mRNA expression for perfusion direction, although tendencies could be observed for higher expression in anterogradely perfused lungs for Ephrin-B2 (Efnb2), endothelial nitric oxide synthase (Nos3), EPH receptor B4 (Ephb4), Krüppel-like-factor2 (Klf2) and prostaglandin I2 receptor (Ptgis) and in retrograde perfusion for vascular cell adhesion molecule 1 (Vcam1) (Figure 9).

## 4. Discussion

Scientific data regarding retrograde flow is rare. Retrograde perfusion could be initiated and maintained for 480 min in IPV and 180 min in IPL. For IPV, a higher perfusion pressure throughout the experiments, as well as increased cytokine levels at the beginning of the experiments could be observed for retrograde perfusion in comparison with anterograde perfusion. In IPL, no differences were observed for perfusion pressure or tidal volume. However, increased edema formation was indicated by increased wet-to-dry ratio and lung weight for retrograde perfusion. Interestingly, no significant differences were found in cytokine levels in perfusate and BALF or mRNA-expression.

For this study, the two set-ups of IPV and IPL were chosen, because pulmonary and systemic vessels differ in several aspects. These differences begin in the basic anatomy and physiology. Pulmonary vessels have a thinner smooth muscle layer compared to systemic vessels [20] and the blood pressure created by the right ventricle is considerably lower than by the left ventricle. Therefore, these two vessel systems can show contrary reactions to the same stimulus. Reversal of flow direction occurs mainly in systemic and only rarely in pulmonary vessels; both arterial and venous systems can be involved. Venous flow reversal is rather common even under physiological circumstances since blood pressure is low in veins and flow is dependent on extremity movement and position. Flow reversal of arterial vessels can only be observed under pathological or iatrogenic conditions. In patients with subclavian steal syndrome retrograde perfusion in vertebral arteries occurs due to a post stenotic pressure drop in a subclavian artery [1]. A flow reversal occurs in the human body while using cardiopulmonary bypass or extracorporeal membrane oxygenators, in case of peripheral arteriovenous cannulation. The cardiopulmonary bypass is often, similar to our experiments, limited to several hours. Furthermore, those procedures challenge the patients’ cardiovascular and immune system. Little is known about the effects of the flow direction on the vessel during those procedures. While these questions could be highly relevant, relatively little is known about the effects of flow reversal in vessels, especially in arteries. Compared to the venous system more serious effects are expected in arteries, where flow reversal only occurs during pathological conditions or iatrogenic interventions. Therefore, the present study focused on arterial IPV to identify potential consequences of retrograde perfusion in an ex vivo setting with physiological conditions where the endothelial cells are embedded in their physiological matrix as opposed to monocultures of endothelial cells on stiff surfaces (GPa range).

Under these conditions, retrograde perfusion in IPV led to a rapid increase in perfusion pressure (indicating a higher vasotonus) and cytokine levels (IL-1β, IL-5, IL-6). The perfusion pressure rose for 7–10 min, and remained at that high level for almost 8 h. Accordingly, Wang and colleagues [1] reported increased phosphorylation of eNOS in bovine aortic endothelial cells caused by 180° flow direction. A decreased endothelial nitric oxide (NO) production could result in increased vasotonus and is physiologically a fast-adapting system. Thus, after the initiation of flow reversal, a sudden drop in NO production with following adaptation and recurrence of NO could be responsible for the increase of perfusion pressure and following steady state in IPV. Accordingly, IL-1β, IL-5 and IL-6 levels are increased and highest after 30 min of retrograde perfusion and then decline over time. In these experiments, there are no increased KC levels at the beginning of the experiments, but rather a rise over the course of the experiment, similar to that described in IPL perfusion [16,21]. IL-1β and IL-5 are mainly secreted from macrophages and T cells [22,23,24]. These cells would be present directly after the beginning of the experiments as adherent or tissue-resident cells [25] with possible consecutive cell death over time explaining the initial increases and following decline of IL-1β and IL-5 levels. In this context, it might be an interesting expansion of the experiment to add immune cells to the perfusate, as IL-1β can have an important impact on vessel permeability [26] and this effect could be even more prominent with the adjusted setup.

Therefore, retrograde perfusion seems to affect rat aorta only at the beginning of flow reversal with following adaptation and no persistent cytokine release or pressure increase. However, most conditions with flow reversal persist after the acute change of perfusion direction stressing the need to investigate long-term consequences. Unfortunately, there was not enough material in IPV to yield sufficient RNA concentrations to perform RT-qPCR. This could be due to the endothelial cells being in the absolute minority compared to the mass of extracellular matrix of the whole blood vessel.

In contrast to the more common flow reversal in systemic vessels, there are only a few circumstances for retrograde perfusion in the pulmonary circulation. Flow reversal in the pulmonary circulation has only been reported in case reports of operative thrombectomy due to pulmonary embolisms [27]. The transfer to the (patho-)physiological conditions of retrograde pulmonary flow in the human body is very limited, since even rare cases, like mitral valve insufficiency, which can be accompanied by a retrograde flow in the pulmonary veins, still differ regarding flow velocity, pressure and disturbances. In experimental settings, studies report edema formation after flow reversal in rats with pulmonary hypertension, if the pulmonary hypertension was caused by vascular endothelial growth factor (VEGF) receptor protein tyrosine kinase 1/2 inhibitor Sugen 5416 but not hypoxia [4]. However retrograde perfusion was performed with increasing flow rates but only for up to 60 min of perfusion and ventilation [4]. Another study from our laboratory observed pulmonary edema formation in retrogradely perfused mouse IPL for 4 h only in combination with an additional hit like acute lung injury [5]. Furthermore, retrograde perfusion led in some mouse IPL to termination of the experiments due to sudden massive pulmonary edema formation. In contrast, in our study, all rat lungs lasted until the end of the experiments, although edema formation was visible and indicated by lung weight increases in retrograde perfusion. Similar to mouse IPL, there was edema formation in rat IPL with retrograde perfusion in our study, although no second trigger was necessary in contrast to the additional acute lung injury in mouse IPL [5]. The reported edema formation could also be due to alterations of NO production described for bovine aortic endothelial cells subjected to 180° flow direction change [6]. Increased NO levels, e.g., via induction of eNOS or iNOS, could cause increased endothelial permeability. Alteration of tight junctions has been shown to be caused by cGMP [28,29,30] and additionally flow reversal could lead to the development of small gaps between adjacent pulmonary endothelial cells [31]. Interestingly, tidal volume and airway resistance seemed not to be affected by progressive edema formation since no significant differences could be reported between anterograde and retrograde perfusion. Furthermore, no significant differences in inflow have been observed in rat IPL with pressure constant retrograde perfusion as opposed to IPV. The presence of a variety of different vessels, such as veins, arteries, arterioles, etc., and different vessel calibers could mask possible effects of altered NO production. Furthermore, if the effects of flow reversal are only present directly after direction change, the pulmonary circulation could compensate for those effects. The setup for the IPL does not only change flow direction but challenges the vessels with a complete pressure gradient as well. In retrograde flow, the pulmonary veins are perfused with the higher precapillary pressure, whereas the pressure of pulmonary arteries is considerably lower, which for both vessel types and their mechano-sensors sets unusual flow characteristics. However, the pulmonary circulation is much more adapted for reactive vasoconstriction or dilation, e.g., to match perfusion to ventilation, known as the Euler–Liljestrand reflex, as the aorta, so in our experiments, both sides, arterial and venous vessels, were able to adapt to the changed conditions, which led to no differences of pressure in comparison. Thus, no differences in mRNA expression could be observed in retrogradely perfused IPL for genes related to vasotonus, such as eNOS, iNOS or endothelin-A receptor (ET-A-R).

Regarding IPL cytokine levels, no significant increases in cytokines could be observed for retrograde perfusion. However, a limitation of the IPL set-up is the rather large volume of perfusion buffer needed to sufficiently fill the system. While IPV required 20 mL of perfusing buffer in a recirculating perfusion due to its lack of a capillary system, IPL needs a volume of 250 mL to function without trapped air. Thus, the rather small increase in cytokines observed in retrograde IPV could simply be diluted in IPL and therefore not traceable. On the other hand, the pulmonary system has several cell types to sense mechanical stress, e.g., endothelial cells, macrophages or type 2 pneumocytes. The IPL setup cannot provide a clear differentiation regarding the responding cells. Cytokines can pass the air–blood barrier, therefore cytokines or chemokines from vascular cells can be found in the alveolar space [32] and cytokines/chemokines from alveolar space will be present in the vascular system [33], especially if this barrier is altered [34]. The PCR measurements include all cell types since whole lung tissue was used. The perfusate samples were most likely influenced by the endothelial cells, but other cell types could also be influential. However, the findings of increased cytokine secretion in the IPV, which is free of any pulmonary cells, supports the role of endothelial cells in inflammation due to flow reversal in our study. Tendencies of initially increased levels in retrograde perfusion could be described for IL-5, and mostly IFN-γ, although no significant differences are visible. BALF cytokine levels in retrograde perfusion can also only be assessed with limitations due to edema formation in IPL. In retrograde IPL, a major part of BALF was pulmonary edema collecting in the alveolae. Thus, determined concentration of cytokines from BALF would be considerably diluted. In this study, no significant increases in cytokine levels could be observed in BALF, although tendencies for increased levels of IL-1β, IL-5 and IL-6 in retrograde IPL are visible. We could assume that without dilution via edema fluid those levels would be considerably higher for retrograde perfusion. One strategy to correct this dilution could be measuring the overall protein concentration in the BALF. However, this approach would not be practical since the edema most likely arose from a permeability change in the pulmonary vascular system. As such the edema would be protein rich [35] and an adjustment according to protein content would not correctly correct the possible dilution of cytokines complicating the comparison between antegrade and retrograde BALF.

## 5. Conclusions

Systemic vessels in vitro (IPV) showed increases in perfusion pressure and cytokine release after initiation of retrograde perfusion. In isolated perfused lungs, retrograde perfusion resulted in progressive edema formation without cytokine release. These observations indicated that retrograde perfusion (1) increases the vasotonus in systemic vessels, (2) causes edema formation in the lungs, and (3) may induce inflammation potentially via emptying of preformed cytokine storage in endothelial cells. Taken together with the current state of the literature these results indicate no particular risks for systemic vessel flow reversal. However, future research needs to focus on long-term set-ups to identify effects exerted by longer flow reversal, e.g., >8 h. For short-term flow reversal in the pulmonary circulation, edema formation is expected and should be monitored. In patients with pre-existing lung conditions, preventive measures should be taken.

## Figures and Tables

**Figure 1 cells-10-01210-f001:**
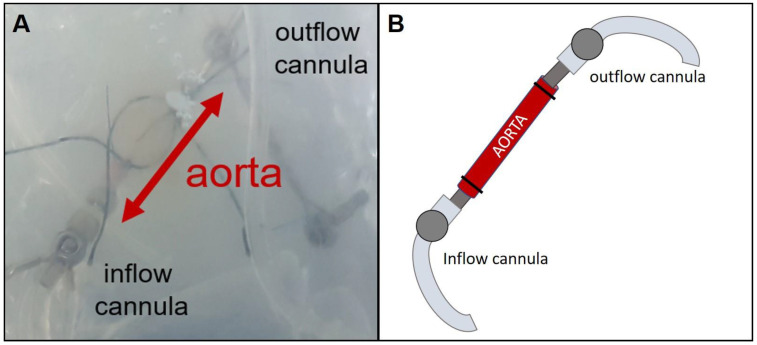
Isolated perfused vessel (IPV) set-up. (**A**): Picture taken through the closed chamber, (**B**)**:** Sketch of the aorta cannulated from both sides allowing perfusion. The aorta was fixed to the cannulas with ligatures.

**Figure 2 cells-10-01210-f002:**
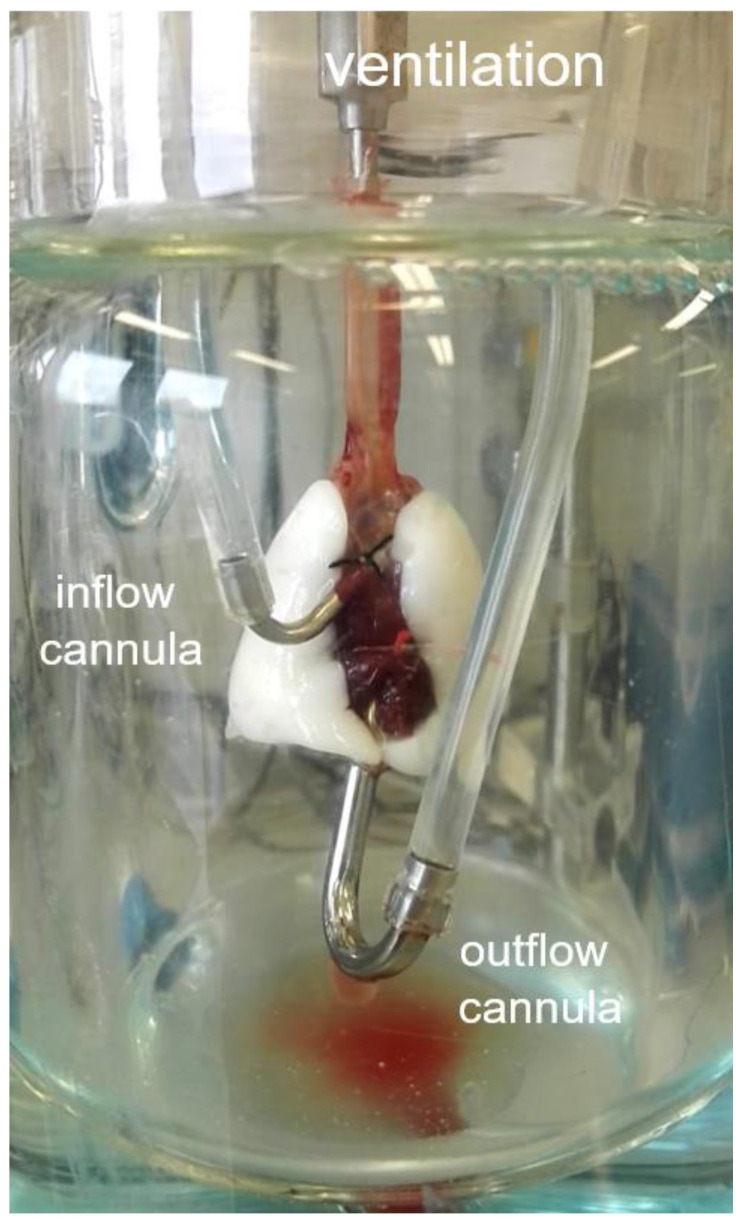
Isolated perfused lungs (IPL) set-up. For anterograde perfusion, the pulmonary artery was cannulated with the inflow cannula and the left atrium was cannulated with the outflow cannula. For retrograde perfusion, the tubings of both cannulas were switched in the cover of the ventilation chamber.

**Figure 3 cells-10-01210-f003:**
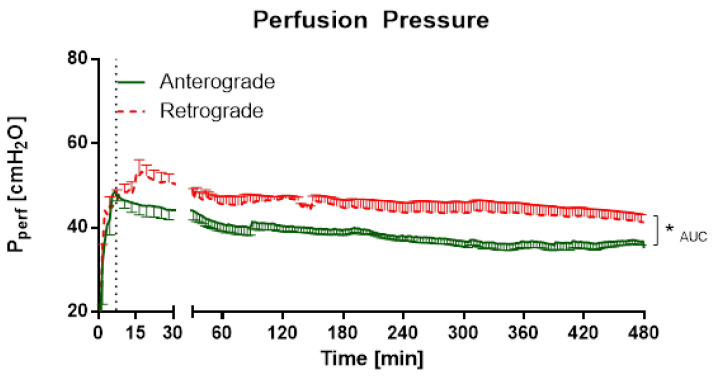
Influence of perfusion direction on perfusion pressure in IPV. The perfusion was established with a step-wise rise of flow up to 30 mL/min. Hydrostatic resistance was added to the outflow by elevating the perfusion buffer reservoir until a perfusion pressure of 45 cm H_2_O was achieved. After that, the hydrostatic resistance and flow were fixed and perfusion pressure was constantly monitored. Mean ± SEM (SEM is pruned to average of 3 min), *n* = 5. * *p* < 0.05.

**Figure 4 cells-10-01210-f004:**
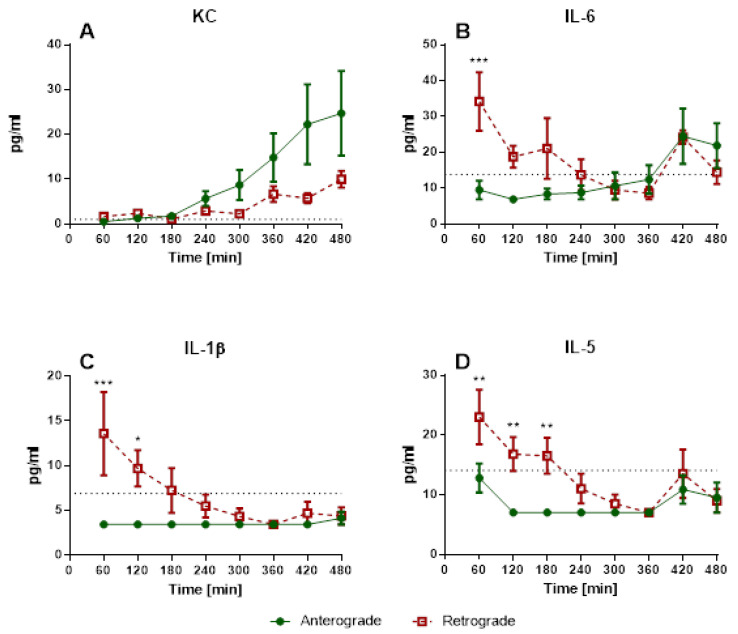
Influence of perfusion direction on perfusate cytokine levels in IPV. Cytokine release of isolated perfused vessels into perfusate. (**A**): KC levels (mean ± SEM), (**B**): IL-6 levels (mean ± SEM), (**C**): IL-1β levels (mean ± SEM), (**D**): IL-5 levels (mean ± SEM), *n* = 5; * *p* < 0.05, ** *p* < 0.01, *** *p* < 0.001 for comparison of anterograde and retrograde for that timepoint. The dotted line indicates the detection limit, samples under the detection limit were assigned the value of half the detection limit.

**Figure 5 cells-10-01210-f005:**
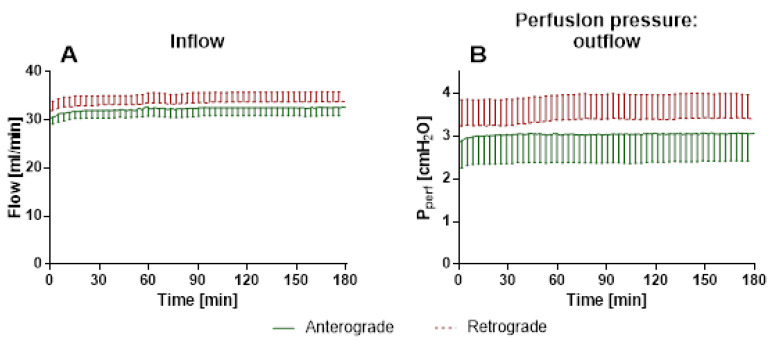
Influence of perfusion direction on physiological parameters in IPL. (**A**): Inflow (mean ± SEM): Due to a pressure-controlled setup, the changes in vasotonus indicate changes in inflow, (**B**): Perfusion pressure (mean ± SEM). Since the inflow is defined, vasotonus changes are shown in the measured pressure. SEM is pruned to an average of 5 min, *n* = 5.

**Figure 6 cells-10-01210-f006:**
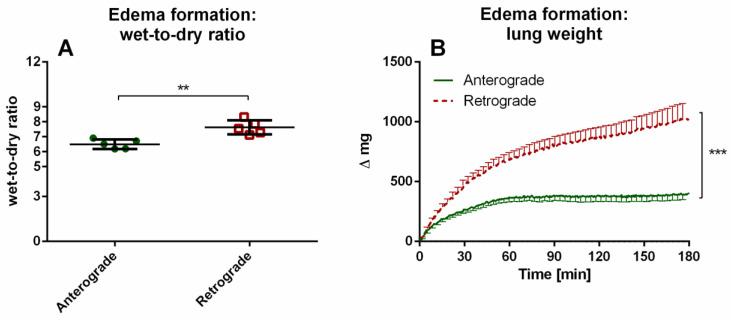
Influence of perfusion direction on edema formation in IPL. (**A**): wet-to-dry ratio (mean ± SEM), (**B**): lung weight (mean ± SEM), SEM is pruned to an average of 5 min, *n* = 5. ** *p* < 0.01, *** *p* < 0.001.

**Figure 7 cells-10-01210-f007:**
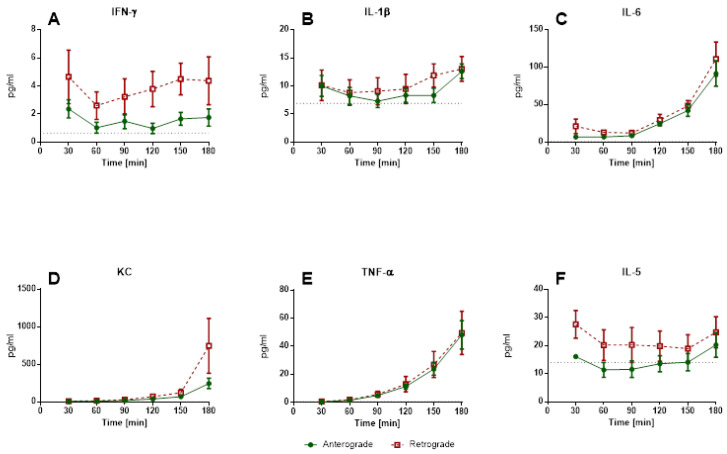
Influence of perfusion direction on perfusate cytokine levels in IPL. Cytokine release of isolated perfused lungs into perfusate. (**A**): IFN-γ (mean ± SEM), (**B**): IL-1β levels (mean ± SEM), (**C**): IL-6 levels (mean ± SEM), (**D**): KC levels (mean ± SEM), (**E**): TNF-α levels (mean ± SEM), (**F**): IL-5 levels (mean ± SEM), *n* = 5. The dotted line indicates the detection limit, samples under the detection limit were assigned the value of half the detection limit.

**Figure 8 cells-10-01210-f008:**
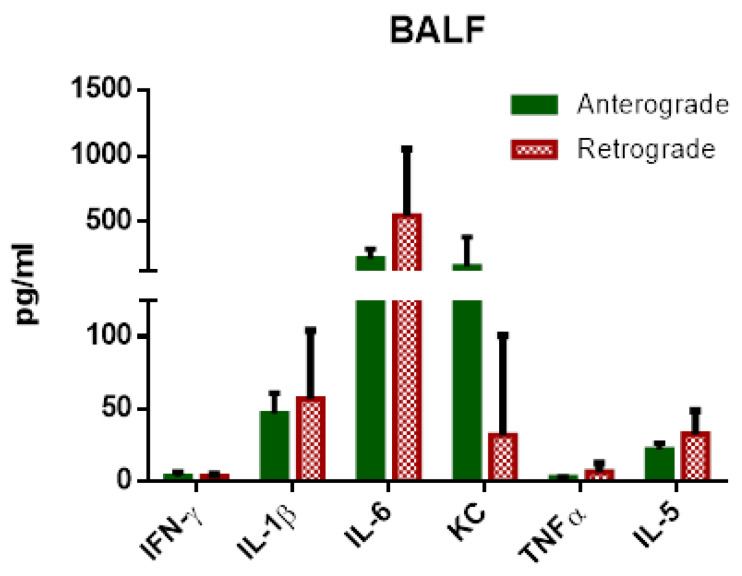
Influence of perfusion direction on BALF cytokine levels in IPL. Cytokine release of isolated perfused lungs into BALF. IFNγ, IL-1β, IL-6, KC, TNF-α and IL-5 levels (mean ± SEM), *n* = 5.

**Figure 9 cells-10-01210-f009:**
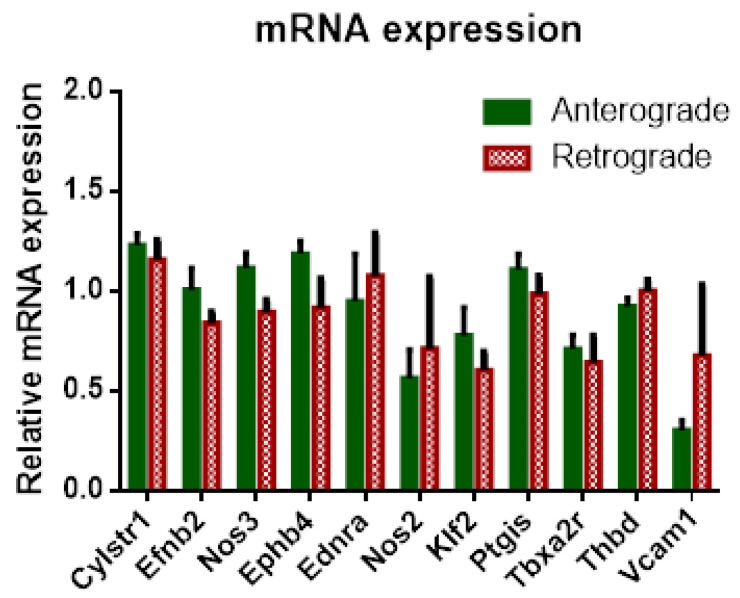
Influence of perfusion direction on mRNA expression in IPL. mRNA expression of cysteinyl leukotriene receptor 1 (CysLt-R1), EphrinB2 (EfnB2), endothelial nitric oxide synthase (Nos3), EPH Receptor B4 (Ephb4), inducible nitric oxide synthase (Nos2) endothelin receptor type A (Ednra), Krüppel-like factor 2 (Klf2), prostaglandin I2 receptor (Ptgis)), thromboxane receptor (Tbx1a2r), thrombomodulin (Tbp) and vascular cell adhesion molecule 1 (Vcam1) (mean ± SEM), *n* = 5.

**Table 1 cells-10-01210-t001:** Primer sequences and annealing temperatures.

Gene	Sequence		Annealing Temperature
Cysltr1	forward:	ACTCCACAATGTACTCTATGATCTC	61 °C
	reverse:	TCACCAAAGAACCACTTGCCT	
Efnb2	forward:	AAGCCAAATCCAGGTTCTAGCA	56 °C
	reverse:	GCGGTACTTGAGCAGCAG	
Nos3	forward:	GATGAATACGATGTGGTATCCCT	60 °C
	reverse:	TTTGTAACTCTTGTGCTGCT	
EphB4	forward:	TGGATGAGAGCGAGAGTTGG	63 °C
	reverse:	GTGCCCGATGAGATATTGCC	
Ednra	forward:	GACAGGTACAGAGCAGTGG	60 °C
	reverse:	GCAGAAGTAGAATCCAAAGAGCC	
Nos2	forward:	GTCTTGGTGAAAGCGGTG	60 °C
	reverse:	AGCAGTAGTTGTTCCTCTTCC	
Klf2	forward:	CTATCTTGCCGTCCTTTGCC	63 °C
	reverse:	CTGTTTAGGTCCTCATCCGT	
Ptgis	forward:	GCCGTGTTATTACTGTTGCTG	60 °C
	reverse:	TAAATATGTCACCGTGCTTCTCCT	
Tbxa2r	forward:	CTGTGAGGTGGAGATGATGG	57 °C
	reverse:	CGGAAGAGGATGTAGACCC	
Thbd	forward:	CGGTCTCAACAGCAACAG	59 °C
	reverse:	CAGGATCTCGGGTATTCAC	
Vcam1	forward:	GTGGACATCTACTCATTCCCT	61 °C
	reverse:	GTAAACATCAGGAGCCAAACAC	
Gapdh	forward:	CGGGGCTCTCCAGAACATCATCC	56 °C
	reverse:	CCAGCCCCAGCGTCAAAGGTG	
Rplp0	forward:	ACAGTACCTGCTCAGAACACC	56 °C
	reverse:	TGCCATTGTCAAACACCTGCT	
Tbp	forward:	TCTTGGCTGTAAACTTGACC	57 °C
	reverse:	CTGGATTGTTCTTCACTCTTGG	

## Data Availability

The datasets generated during and/or analyzed during the current study are available from the corresponding author on reasonable request.

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
