# Peer review of "Never Change a Flowing System? The Effects of Retrograde Flow on Isolated Perfused Lungs and Vessels"

_cells, 2021, doi:10.3390/cells10051210_

Round 1

Reviewer 1 Report

Krabbe H et al investigated the effect of retrograde perfusion on endothelial cell damage in aorta and lungs ex vivo, based on the hypothesis that the retrograde flow may cause endothelial dysfunction. I have several points they have to clarify.

  1. Abstract. I do not understand what the following sentence means: “In IPL, increased edema….though dilution by for example pulmonary edema cannot be excluded.” Thye should revise.
  2. I disagree with this sentence “Therapeutically, the cardio¬pulmonary bypass and extracorporeal membrane oxygenation …. are based on a constant retrograde flow,…” The situation of retrograde perfusion on ECMO or CPB is only at the V-A perfusion with peripheral cannulation, and a lot of cases are used as VV-ECMO and central cannulation of CPB creating antegrade perfusion. They should revise this.
  3. Method 2.4 “…or retrogradely via left ventricle and right pulmonary artery (Fig. 2)…”. How did they take care of the mitral valve? Also return flow should come back from both PA.
  4. They used only 2% BSA as colloid material in perfusate and this is quite low to keep pulmonary hydrostatic balance. At least, 4-5% of BSA is required for keeping the balance. Also, it is not clear if they included calcium ion in perfusate.
  5. In 2.6, they listed 3 genes for internal control for qPCR, but they did not describe which gene was used to normalize their target genes. They should clarify.
  6. They showed the gene expression of “Etar” which is not specified in anywhere. Based on sequence, the primers can detect endothelin receptor type A (Ednra). Please clarify.
  7. Their providing primer sequence for Ptgir can detect prostaglandin I2 synthase (Ptgis). Please check the primer sequence.
  8. They analyzed the normality of the data distribution but did not perform statistical analysis to compare the difference between the groups after that. 2-way ANOVA with posthoc test for analyzing data with time-dependent change and a test to compare 2 groups are required.
  9. Section 3.1.2. It is not clear what the resistance for. Vascular resistance? Airway resistance?
  10. This is not the first report to test the effect of retrograde perfusion. It does not matter >60min or animal species. They should tone down.
  11. The overall discussion contains a lot of aside stories. Also they discuss a lot with expectation and hopes. (e.g. Weibel-Palade bodies, macrophage, T-cells, bronchial artery, NO synthesis etc.) They should revise the entire discussion based on their findings objectively.
  12. In discussion 2nd paragraph, “In a physiological context, …in oxygen levels with vasoconstriction.” Sounds like the pulmonary vasculature does not respond to hypoxia. This is not true, hypoxic pulmonary vasoconstriction exists and it is reactive to oxygen content. They should remove this sentence.
  13. The following sentences in the discussion are not appropriate and do not make sense. Please remove them. “In addition to that, physicians can induce a flow reversal…One factor contributing to this could be the retrograde flow.”
  14. Rather than IL-6, IFN-g had a bigger difference between the groups. They should revise the discussion accordingly.
  15. Mice do have bronchial arteries. In Ref29, the authors were not able to find it by microsphere injection because of particle size. The related sentence could mislead the readership so should be removed or modified.

Reviewer 2 Report

The paper by Krabbe and colleagues shows a descriptive analysis of the inflammatory cytokines released by isolated lungs and aorta in response to retrograde or anterograde flow. This may be of relevance during surgical procedures or during conditions such as subclavian steal syndrome.

Major concerns

  1. In the conditions in which the IPL is performed, hydrostatic pressure reaches 45cmH2O. This is known to induce edema development. How do the authors dissect the percentage of edema that is deriving from increased pressure and not flow (retrograde or anterograde)
  2. The authors changed the start position of perfusion to study anterograde vs retrograde flow. The mechano-sensing proteins in the vessel wall and across the pulmonary circulation are differently distributed. How do the authors approach the compensatory mechanisms that occurs via the temporal activation of different signaling pathways? This should be approached in the discussion
  3. In the same line, retrograde perfusion in isolated lungs do not mimic the clinical conditions of retrograde flow.
  4. What is the rationale for the use of only female rats?
  5. Figure 6A sow a highly significant statistical differenceand it seems like the values are not much different between groups. How was this analysis performed?
  6. Authors conclude that retrograde flow improve vasotonus in systemic vessels: please explain how this conclusion is possible as there are no data on vascular tone
  7. Authors should take in consideration that other cell types in the lung are key for inflammatory response (specially alveolar type II cells). Due to the close proximity of lung capillaries and epithelial barrier is is likely that epithelial cells are responding to mechanical stimuli in the lungs. How do authors conclude the the inflammatory response is driven by the endothelial cells?

Minor:

  1. Authors should describe how lung weight is monitored in real-time during the IPL
  2.  Why the aortic arch was not used since this is where baroceptors are heavily expressed?

Round 2

Reviewer 2 Report

The authors addressed my main technical concern in regard to the perfusion pressure of the lungs. Some of their responses are limited and justification for the use of only females is not scientifically sound. They could make an extra effort to improve the discussion with the topics pointed on the first round of the review. Adding a sentence "On the other hand, the pulmonary
system has several cell types, e.g. type 2 pneumo-cytes, which could react to changes in blood flow."  does not improve the discussion significantly.  It is acceptable to use the thoracic aorta as they have pointed out that the baroceptor response is lost in an isolated vessel preparation. 
